# Linear EUS Accuracy in Preoperative Staging of Gastric Cancer: A Retrospective Multicenter Study

**DOI:** 10.3390/diagnostics13111842

**Published:** 2023-05-25

**Authors:** Germana de Nucci, Tommaso Gabbani, Giovanna Impellizzeri, Simona Deiana, Paolo Biancheri, Laura Ottaviani, Leonardo Frazzoni, Enzo Domenico Mandelli, Paola Soriani, Maurizio Vecchi, Gianpiero Manes, Mauro Manno

**Affiliations:** 1Gastroenterology and Digestive Endoscopy Unit, ASST Rhodense, 20094 Garbagnate Milanese, Italy; 2Gastroenterology and Digestive Endoscopy Unit, Azienda USL Modena, 41012 Carpi, Italym.manno@ausl.mo.it (M.M.); 3Gastroenterology Unit, Surgical and Medical Sciences Department, Sant’Orsola Malpighi Hospital, 40138 Bologna, Italy; 4Gastroenterology Unit, Major Policlinic Hospital, 20122 Milan, Italy

**Keywords:** gastric cancer staging, linear endoscopic ultrasound, diagnostic accuracy

## Abstract

Introduction: Preoperative gastric cancer (GC) staging is the most reliable prognostic factor that affects therapeutic strategies. Contrast-enhanced computed tomography (CECT) and radial endoscopic ultrasound (R-EUS) scans are the most commonly used staging tools for GC. The accuracy of linear EUS (L-EUS) in this setting is still controversial. The aim of this retrospective multicenter study was to evaluate the accuracy of L-EUS and CECT in preoperative GC staging, with regards to depth of tumor invasion (T staging) and nodal involvement (N staging). Materials and methods: 191 consecutive patients who underwent surgical resection for GC were retrospectively enrolled. Preoperative staging had been performed using both L-EUS and CECT, and the results were compared to postoperative staging by histopathologic analysis of surgical specimens. Results: L-EUS diagnostic accuracy for depth of invasion of the GC was 100%, 60%, 74%, and 80% for T1, T2, T3, and T4, respectively. CECT accuracy for T staging was 78%, 55%, 45%, and 10% for T1, T2, T3, and T4, respectively. L-EUS diagnostic accuracy for N staging of GC was 85%, significantly higher than CECT accuracy (61%). Conclusions: Our data suggest that L-EUS has a higher accuracy than CECT in preoperative T and N staging of GC.

## 1. Introduction

Gastric cancer (GC) is the fifth most common cancer and the third most common cause of cancer-related death globally. Risk factors for GC include *Helicobacter pylori* infection, age, high dietary intake of salt, and low dietary intake of fruits and vegetables. GC is usually diagnosed histologically on endoscopic biopsies. Preoperative GC staging involves contrast-enhanced computed tomography (CECT), which is currently considered the “gold standard” test, and may involve endoscopic ultrasound (EUS) and positron emission tomography (PET) scans. Preoperative GC staging is the most accurate prognostic factor in predicting surgical outcome and 5-year survival rate [1]. Furthermore, accurate GC staging allows planning the best treatment approach, thereby limiting unnecessary exploratory surgeries.

Radial endoscopic ultrasound (R-EUS) has been shown to be a reliable non-surgical diagnostic test for GC staging. Compared to linear EUS (L-EUS), R-EUS provides images that are easier to interpret [2]. Nevertheless, the use of L-EUS in GC staging has increased in the last few years. Compared to R-EUS, L-EUS has several advantages, such as clearer visualization of cardial neoplasms and better evaluation of esophageal involvement [3]. Furthermore, L-EUS provides better identification of suspected malignant lymph nodes and allows for performing fine needle aspiration (FNA) or fine needle biopsy (FNB) for confirmation of malignancy [4]. L-EUS accuracy in GC staging, however, is still controversial and needs further validation.

In this retrospective multicenter study, we evaluated the accuracy of L-EUS and CECT in preoperative GC staging with regards to depth of tumor invasion (T staging) and lymph node involvement (N staging) by comparing their results with postoperative histopathologic GC staging.

## 2. Materials and Methods

### 2.1. Patients

We enrolled 191 consecutive patients (113 men and 78 women; age range 38–91 years; mean age 66 years) who underwent preoperative GC staging at AUSL Modena (Carpi Hospital, Modena, Italy) and ASST Rhodense (Garbagnate Milanese Hospital, Milan, Italy) EUS services between April 2017 and April 2020. Demographic, clinical, histopathologic, EUS, and radiologic data were collected retrospectively from medical notes and endoscopic and radiologic databases and recorded in a common database. Exclusion criteria were age < 18 years, current pregnancy, other primary synchronous cancers, previous or planned neoadjuvant therapy, and severe organ failure.

### 2.2. Study Design

All the enrolled patients had a firm diagnosis of GC, made by esophagogastroduodenoscopy and histology, and, after completing preoperative staging with L-EUS and CETC, they underwent upfront resective surgery. Preoperative TNM staging, according to the 8th edition of the AJCC manual, was performed by L-EUS and CECT of the chest, abdomen, and pelvis, aimed at assessing tumor depth of invasion (T staging), lymph node involvement (N staging), and metastatic (M staging) disease. Post-surgical staging was subsequently performed by histopathologic examination of surgical specimens by expert pathologists not aware of EUS and CECT staging results. The accuracy of L-EUS and CECT preoperative GC staging was subsequently calculated by comparing the results of these techniques with postoperative histopathologic GC staging. The procedures of this study are in accordance with the Helsinki Declaration of 1975, as revised in 2000. This study has been approved by the Ethics Committee of the Area Vasta Emilia Nord, Italy.

### 2.3. L-EUS

Two different types of linear echoendoscopes were used, i.e., the Olympus GF-UCT180 and the Pentax EG-3870UTK/EG-3270UK Slim. All L-EUS procedures were performed by expert echoendoscopists (GDN, TG, EDM), who had been performing > 500 EUS exams/year in the previous 3 years. All L-EUS exams were performed in the same manner with the patient in left lateral decubitus, under conscious or deep sedation, and by insufflation of carbon dioxide. The frequency of ultrasound waves was set from 10 to 12 MHz. The echoendoscope was advanced into the gastric lumen, and the lesion was first examined endoscopically. Instillation of 0.9% isotonic saline solution was routinely performed into the gastric lumen in order to submerge the lesion and obtain better L-EUS images (water-filling technique). The stomach was studied starting from the pylorus to the cardia in order to study cancer-related wall abnormalities and lymph node involvement. Our findings were recorded in a database and interpreted following a standard protocol according to the normal 5-layered endosonographic structure of the gastric wall. GCs were staged according to the TNM classification criteria of the International Union for Cancer (UICC). Echoendoscopic staging of the depth of invasion (uT staging) was performed by assessing which of the normal 5 layers of the gastric wall structure were altered by the hypoechoic neoplastic lesion [5]. Echoendoscopically, lesions involving one of the first three layers of the gastric wall (neoplasms infiltrating the mucosa, the muscularis mucosae, or the submucosa) were classified as uT1. Lesions invading the fourth layer of the gastric wall (neoplasms infiltrating the muscularis propria) were defined as uT2, while lesions showing evidence of sub-serosal involvement were classified as uT3. Finally, lesions showing signs of invasion beyond the serosa or to adjacent organs and structures were defined as uT4. The uT1 stage was indicative of early GC, while the uT2, uT3, and uT4 stages were indicative of advanced GC. Regarding echoendoscopic evaluation of lymph node involvement (uN staging), the following nodal stations were assessed: perigastric greater and lesser curvature nodes; peripancreatic nodes, splenic hilum and splenic vascular axis nodes; hepatic hilum and portal vein nodes; celiac trunk nodes; mesenteric vascular axis nodes; paraaortic and periesophageal nodes; infradiaphragmatic and supradiaphragmatic nodes. Lymph nodes >10 mm in diameter with a rounded shape and a hypoechoic echo-structure with vascular hilum disappearance were considered malignant [6]. The absence of involved lymph nodes was defined as uN0, whereas uN+ was used to describe the presence of involved lymph nodes.

### 2.4. CECT

Radiologic preoperative TNM GC staging, according to the 8th edition of the AJCC manual, was performed by CECT of the chest, abdomen, and pelvis. Staging CT was performed using the standard protocol for gastric cancer suggested by Italian Society of Radiology guidelines (Documeno SIRM 2022. Protocolli di Tomografia Computerizzata per Indicazione Clinica. September 2022 [ISBN (E-BOOK) 979-12-80086-61-7]), in accordance with the European Society of Radiology guidelines. With regards to radiologic assessment of depth of invasion (T staging), GCs characterized by focal thickening of the inner layer of the gastric wall, with recognizable outer layers, and with a clear fatty layer around the gastric wall were described as T1. GCs characterized by focal or diffuse thickening of the gastric wall with transmural involvement and regular external wall margins were defined as T2. GCs with involvement of the external gastric wall margins were defined as T3. GCs with serosal layer invasion or obliteration of the adipose plane between the neoplasm and the adjacent organs were defined as T4.

### 2.5. Histopathology

Postoperative GC staging on the surgical specimens was performed by expert pathologists not aware of EUS and CECT staging results. Histopathologic GC staging was formulated according to the TNM classification criteria, in keeping with the 8th edition of the AJCC manual. With regards to histopathologic assessment of depth of invasion (pT staging), neoplasms were classified as pT1 when there was invasion of the lamina propria, muscularis mucosae, or submucosa; pT2 when there was invasion of the muscularis propria; pT3 when the serosa was invaded; and pT4 when there was invasion beyond the serosa or to the adjacent organs and structures. Regarding pathologic evaluation of lymph node involvement (pN staging), the absence of involved lymph nodes was defined as pN0, whereas the presence of involved lymph nodes was classified as pN+.

### 2.6. Statistical Analysis

Preoperative GC T staging and N staging by EUS and CECT were compared with postoperative histopathologic staging of surgical specimens. The sensitivity, specificity, and accuracy of both EUS and CECT T staging and N staging were calculated. Positive likelihood ratios (LR+) and negative likelihood ratios (LR−) can be used to assess the performance of a diagnostic test. LR+ >10 and LR− <0.1 suggest that the test has a high probability of correctly confirming or excluding a certain condition, respectively. With regards to N staging, EUS and CECT’s positive predictive value and negative predictive value, as well as their LR+ and LR−, were determined. Receiver operating characteristic (ROC) curve analysis was used to evaluate the overall concordance between EUS/CECT and histopathologic staging. Our database was completed using Office Excel 2007. Statistical analysis was performed using STATA version 13 (Stata Corp., College Station, TX, USA).

## 3. Results

Demographic data of the study population (*n* = 191) and characteristics of GCs are summarized in Table 1. Upon endoscopic evaluation, 95 lesions (49%) were ulcerated, 19 (9%) were vegetating, 67 (35%) were mixed (both vegetating and ulcerated), and in 10 cases (7%) the GC presented as linitis plastica. Localization of the GC was cardia in 2 cases (2%), fundus in 12 cases (6%), gastric body in 79 cases (41%), angulus in 32 cases (16%), antrum in 65 cases (34%), and pylorus in 1 case (1%). In 15 cases (7%), the neoplasm had a circumferential extension with a tendency to progressively induce stenosis of the gastric lumen; in 9 cases (5%), the neoplasm involved the greater curvature of the stomach; in 64 (33%) the lesser one; in 52 (28%) the anterior wall; and in 51 (27%) the posterior wall.

### 3.1. Histopathologic Characteristics of GCs

Tumor histotypes were distributed as follows: well differentiated in 35 patients (18%), moderately differentiated in 47 patients (24%), and poorly differentiated in 109 patients (58%). According to Lauren’s classification, 69% of cases (131) were diffuse-type tumor, and 31% of cases (60) were intestinal-type tumors. The grade of differentiation was distributed as follows: G1 had 37 cases, G2 had 84 cases, G3 had 63 cases, and G4 had 7 cases. Regarding the type of surgery performed, 94 patients (49%) underwent total gastrectomy, while 97 (51%) underwent partial gastrectomy. From a histopathologic point of view, 9 (5%) patients presented with early GC (pT1), while 182 (95%) presented with advanced disease (pT2, pT3, or pT4). Out of 191 patients, 123 (64%) had lymph node involvement (pN+) (Table 1).

### 3.2. Depth of Invasion (T Staging)

As displayed in Table 2, L-EUS diagnostic accuracy in T staging was 100% for T1, 60% for T2, 74% for T3, and 80% for T4 GCs. Out of 191 GCs, 140 were correctly staged with regards to the depth of gastric wall invasion, and the overall diagnostic accuracy in T staging was 73% (95% CI 66–79). Out of 191 GCs, 28 (15%) were overstaged by EUS, while 23 (12%) were understaged by L-EUS. Among the overstaged GCs, 17 were pT2 and 11 were pT3, whereas among understaged GCs, 2 were pT2, 12 were pT3, and 9 were pT4. GC characteristics most often associated with L-EUS T staging mistakes were: antrum anterior wall localization (29%), lesion size > 4 cm (56%), and mixed endoscopic appearance (both vegetating and ulcerated) of the lesion. In particular, 45% of GCs with mixed endoscopic appearance were incorrectly staged by L-EUS, as compared to 33% of ulcerated GCs and 27% of vegetating GCs. The mean tumor size of the 140 GCs for which L-EUS provided correct T staging was 4.1 cm, and this was significantly (*p* < 0.01) smaller than the mean tumor size (4.9 cm) of the 51 GCs for which T staging by L-EUS was incorrect. The histotype of GCs for which L-EUS provided correct T staging did not differ significantly compared to incorrectly staged GCs.

As shown in Table 3, CECT diagnostic accuracy in T staging was 78% for T1, 55% for T2, 45% for T3, and 10% for T4 GCs. Only 77 out of 191 GCs were correctly staged with regards to the depth of gastric wall invasion, and the overall diagnostic accuracy in T staging was 40% (95% CI 33–47). Out of 191 GCs, 21 (11%) were overstaged, while 93 (49%) were understaged by CECT. Among the GCs that were overstaged by CECT, there were 16 pT2 and 5 pT3 GCs. Among the GCs understaged by CECT, there were 2 pT1, 5 pT2, 44 pT3, and 42 pT4 GCs. GCs with mixed endoscopic appearance (both vegetating and ulcerated), ulcerated GCs, and vegetating GCs accounted for 38%, 47%, and 25% of CECT T staging errors, respectively.

### 3.3. Lymph Node Involvement (N Staging)

As displayed in Table 4, L-EUS provided correct preoperative N staging in 164 out of 191 (85%) GCs. L-EUS correctly diagnosed N0 disease in 65 out of 68 (95%) patients with no lymph node involvement at the post-surgical examination of the surgical specimens, whereas L-EUS overstaged (false positive result) lymph node involvement in 3 cases (5% of pN0 GCs). Out of the 123 GCs with lymph node involvement (pN+), L-EUS correctly diagnosed 99 cases (80% of pN+ GCs), while in 24 cases (20% of pN+ GCs), lymph node involvement was understaged (false negative result) by L-EUS.

CECT provided correct preoperative N staging in 118 out of 191 (61%) GCs (Table 4). CECT correctly diagnosed N0 disease in 64 out of 68 (94%) patients with no lymph node involvement at the post-surgical examination of the surgical specimens, whereas CECT overstaged (false positive result) lymph node involvement in 4 cases (6% of pN0 GCs). Out of the 123 GCs with lymph node involvement (pN+), CECT correctly diagnosed 54 cases (43% of pN+ GCs), while in 69 cases (57% of pN+ GCs), lymph node involvement was understaged (false negative result) by CECT.

As shown in Table 5, with regards to preoperative N staging of GCs, L-EUS showed an overall accuracy of 85% (95% CI 80–90), a sensitivity of 80% (95% CI 72–87), and a specificity of 95% (95% CI 87–99). L-EUS positive and negative predictive values in diagnosing N+ GCs were 97% (95% CI 91–99) and 73% (95% CI 62–81), respectively, with a LR+ of 18.20 (95% CI 6–55) and a LR− of 0.20 (95% CI 0.14–0.29). Regarding preoperative N staging of GCs, CECT displayed an overall accuracy of 61% (95% CI 54–68), a sensitivity of 43% (95% CI 35–53), and a specificity of 94% (95% CI 85–98). CECT positive and negative predictive values in diagnosing N+ GCs were 93% (95% CI 83–98) and 48% (95% CI 39–56), respectively, with a LR+ of 7.46 (95% CI 2.83–19.7) and a LR− of 0.59 (95% CI 0.50–0.70). Compared to CECT, L-EUS showed significantly (*p* < 0.05) higher overall accuracy, sensitivity, and negative predictive value and a significantly lower LR− in correctly performing preoperative N staging of GCs (Table 5).

L-EUS and CECT’s overall diagnostic accuracy in preoperative N staging of GCs can be compared by evaluating the area under the ROC curve (AUC) of each test. In this setting, L-EUS showed significantly (*p* < 0.005) higher accuracy than CECT, with an AUC of 0.88 (95% CI 0.83–0.92), as compared with 0.69 (95% CI 0.63–0.74), respectively (Figure 1).

## 4. Discussion

Preoperative staging is the most important factor for establishing patient management (upfront surgery vs. neoadjuvant or palliative treatment) and determining prognosis in GC. EUS accuracy in GC staging has been previously evaluated in several studies [3,4,5,6,7,8,9,10,11,12,13,14,15,16,17,18,19,20], and it ranges between 65% and 92% for T staging, and between 50% and 90% for N staging. Most of the published papers on preoperative GC staging by EUS have been performed using R-EUS. Here, we have assessed the diagnostic performance of L-EUS. This latter has been shown to be superior to R-EUS in the diagnosis of pancreatic and biliary disease; however, the use of L-EUS for gastrointestinal luminal wall lesions is currently not widespread, and its diagnostic performance in this setting has not been fully determined yet. Staging with R-EUS has the advantage of allowing for 100% of the visceral circumference in a single view. With L-EUS, it is necessary to perform a site-by-site sectoral assessment. However, this could allow for more accurate lymph node staging, especially in the more distant stations, which are not directly close to the lesion and therefore are configured as metastatic, allowing FNB to be performed.

Our data show that L-EUS has an overall diagnostic accuracy of 73% in T staging of GCs, while L-EUS provided incorrect T-staging in 27% of cases (due to overstaging in 15% of cases and understaging in 12% of cases). L-EUS shows better diagnostic performance in pT1, pT3 and pT4 GCs, whereas it may overstage pT2 GCs, which involve the muscular layer [8,21,22,23,24,25,26]. The possible reasons for the relatively low performance of L-EUS in correctly staging pT2 GCs may be connected to its relatively poor ability to distinguish between the *muscularis propria* and the serosal layer or to the absence of the serosal layer in some stomach portions (lesser curve, posterior wall of the gastric fundus, anterior wall of the gastric antrum) [1]. Accordingly, our data show that the performance of L-EUS in T staging was lower for pT2 GCs localized on the anterior wall of the gastric antrum. Furthermore, poorly differentiated histotype and larger tumor size have both been described as factors associated with inaccurate T-staging of GC by L-EUS [27]. Our data confirm that the performance of L-EUS for T staging of GCs was significantly better for smaller lesions, whereas in our study, the ability of L-EUS to provide correct T staging of GCs was not significantly affected by the histologic degree of differentiation.

In our study, we observed that CECT had a GC T staging overall accuracy of only 40% (95% CI 33–47), which is lower than that reported in previous studies, where it ranged between 71.4% and 88.9% [28,29,30,31]. This difference could be attributed to the lack of dedicated CECT protocols in countries with a lower incidence of GC, including Italy, which could especially affect CECT sensitivity. Furthermore, we observed that post-surgical analysis of surgical specimens revealed a higher T staging compared to CECT T staging in 49% of patients, suggesting that CECT may have a tendency to understage GC. Among the GCs understaged by CECT, the large majority were pT3 and pT4 lesions, which strongly suggests a tendency for CECT to understage locally advanced disease. Ulcerated GCs accounted for 47% of CECT T staging errors, whereas errors in T-staging of this type of lesion were not particularly frequent with L-EUS.

With regards to preoperative N staging of GCs, L-EUS provided correct staging in 85% of cases. Among GCs for which N staging by L-EUS was incorrect, L-EUS showed mainly a tendency to understage lymph node involvement (20% of false negatives out of 123 patients with pN+ GC). This is likely related to the lack of definitive endosonographic criteria to distinguish inflammatory from metastatic lymph nodes. Moreover, in doubtful cases (*n* = 12/191), it was preferred to hypothesize positive lymph node involvement. This is because, in any case, it would not have changed the type of therapy, while the execution of the FNA/FNB could have determined a non-negligible risk of seeding. CECT provided correct N staging in only 61% of cases, which is slightly lower than the 64–78% previously reported in the literature [28,29,30,31]. Furthermore, CECT showed a higher rate of false-negative N staging compared to L-EUS (57% vs. 20%, respectively). In our study, the comparison between the Area Under the ROC Curves of L-EUS and CECT (0.88 vs. 0.69, respectively) highlighted a significantly higher overall accuracy of L-EUS in the N staging of GCs. The specificity and positive predictive value of L-EUS (95% and 97%, respectively) and CECT (94% and 93%, respectively) in N staging of GC were both high and comparable. We observed that L-EUS has a significantly higher accuracy, sensitivity, and negative predictive value than CECT in correctly performing preoperative N staging of GCs. The LR+ value of 18.20 observed for N staging strongly suggests that L-EUS is a conclusive diagnostic test for correctly identifying N+ GCs. Conversely, CECT has a LR+ value < 10 (7.46) for correctly diagnosing N+ GCs, which, albeit not significant, suggests that CECT is a less conclusive test than L-EUS in predicting the presence of lymph node involvement in GC. The L-EUS LR− value for N staging of GC was 0.20, indicating a moderately useful test in predicting the absence of lymph node involvement in GC. The CECT LR− value for N staging of GC was 0.59, which is significantly higher than the L-EUS LR−, indicating poor utility of CECT in predicting the absence of lymph node involvement in GC due to a high number of false negative results. Taken together, our data show that L-EUS is able to provide essential preoperative information about lymph node involvement in GC. This, in turn, has an important impact on patient management by guiding the decision between upfront surgery (when the N stage is uN0) or neoadjuvant therapy (in the case of uN+ GC).

The main limitations of our study are its retrospective design and the fact that L-EUS procedures, which are operator-dependent, had been performed by three different operators, although these were expert echoendoscopists who had been performing > 500 EUS exams/year in the previous 3 years.

Finally, and importantly, our data about L-EUS accuracy in preoperative GC staging are comparable to those previously reported in the literature about R-EUS performance in the same setting. With regards to T staging of GCs, the accuracy of L-EUS that we observed in our study is not inferior to the data previously reported for R-EUS [27]. Regarding N staging of GC, it has been reported that the accuracy of R-EUS is 80% [3], slightly lower than the overall accuracy of L-EUS (85%), which we observed in our study.

## 5. Conclusions

Our study shows that L-EUS is an accurate and valuable diagnostic tool for preoperative GC staging. In particular, L-EUS provides accurate information about both the depth of invasion and lymph node involvement in GC. We observed that L-EUS has a significantly higher accuracy than CETC in preoperative N staging of GC. Our data also suggest that L-EUS accuracy in GC staging is similar to R-EUS. Compared to R-EUS, in selected cases, L-EUS has the advantage of allowing FNA or FNB of suspicious lymph node metastases, thereby improving the accuracy of preoperative staging and, consequently, patient therapy management and prognosis.

## Figures and Tables

**Figure 1 diagnostics-13-01842-f001:**
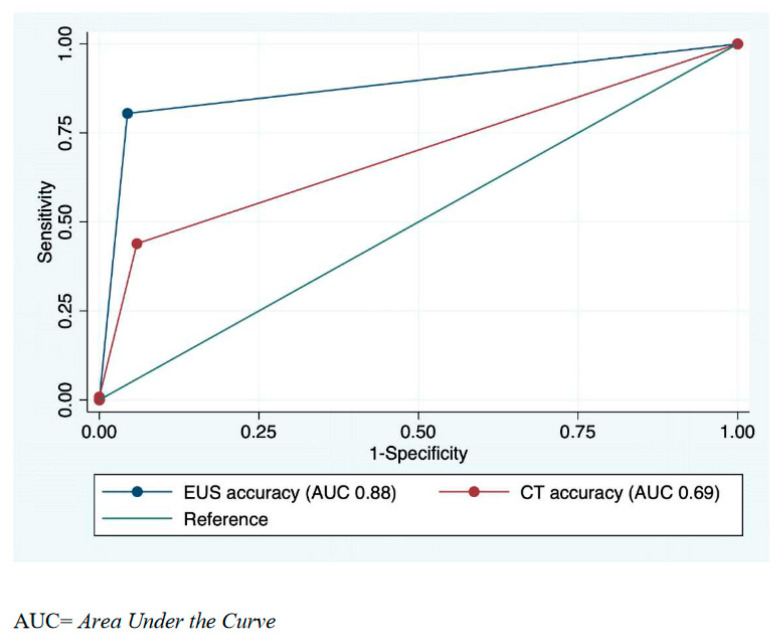
Comparison of linear EUS and CECT ROC curves in preoperative N staging of gastric cancer.

**Table 1 diagnostics-13-01842-t001:** Demographic characteristics of the study population (*n* = 191) and gastric cancer features.

Age (mean, range)	66 (38–91)
Sex (male/female)	113/78
Tumor maximum diameter (cm) (mean, range)	4.3 (1–8)
Histotype (well/moderately/poorly differentiated)	35/47/109
Lauren Classification (diffuse/intestinal) Grade of differentiation (G1/G2/G3/G4) Stage pT (T1/T2/T3/T4)	131/60 37/84/63/7 9/47/88/47
Stage pN (pN0/pN+)	68/123

**Table 2 diagnostics-13-01842-t002:** Linear EUS accuracy in preoperative T staging of gastric cancer.

Histopathologic Staging	* n *	Correct L-EUS Staging *n*/% (95% CI)	L-EUS Overstaging *n*/% of Correct Staging/L-EUS Stage	L-EUS Understaging *n* /% of Correct Staging/L-EUS Stage
pT1	9	9/100 (66–100%)	-	-
pT2	47	28/60 (44–73%)	17/36/uT3	2/4/uT1
pT3	88	65/74 (63–82%)	11/12/uT4	12/14/uT2
pT4	47	38/80 (66–90%)	-	9/20/uT3
Total	191	140/73 (66–79%)	28/15	23/12

**Table 3 diagnostics-13-01842-t003:** CECT accuracy in preoperative T staging of gastric cancer.

Histopathologic Staging	* n *	Correct CECT Staging *n*/% (95% CI)	CECT Overstaging *n*/% of Correct Staging	CECT Understaging *n*/% of Correct Staging
pT1	9	7/78 (39–97%)	-	2/22
pT2	47	26/55 (40–69%)	16/34	5/11
pT3	88	39/45 (33–55%)	5/5	44/50
pT4	47	5/10 (3–23%)	-	42/90
Total	191	77/40 (33–47%)	21/11	93/49

**Table 4 diagnostics-13-01842-t004:** Linear EUS and CECT accuracy in preoperative N staging of gastric cancer.

Histopathologic Staging	* n *	Correct L-EUS Staging *n*/%	Correct CECT Staging *n*/%	Incorrect L-EUS Staging *n*/%	Incorrect CECT Staging *n*/%
pN0	68	65/95	64/94	3/5	4/6
pN+	123	99/80	54/43	24/20	69/57
Total	191	164/85	118/61	27/15	73/39

**Table 5 diagnostics-13-01842-t005:** Linear EUS and CECT diagnostic performance data in preoperative N staging of gastric cancer.

Diagnostic Performance Parameters	L-EUS % (95% CI)	CECT % (95% CI)
Accuracy *****	85% (80–90%)	61% (54–68%)
Sensitivity *****	80% (72–87%)	43% (35–53%)
Specificity	95% (87–99%)	94% (85–98%)
PPV	97% (91–99%)	93% (83–98%)
NPV *****	73% (62–81%)	48% (39–56%)
LR+	18.20 (6.01–55.3)	7.46 (2.83–19.7)
LR- *****	0.20 (0.14–0.29)	0.59 (0.50–0.70)

***** statistically significant difference between linear EUS and CECT; LR+, positive likelihood ratio; LR-, negative likelihood ratio; NPV, negative predictive value; PPV, predictive positive value.

## Data Availability

Data available on request due to restrictions (privacy). The data presented in this study are available on request from the corresponding author.

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
