# Peer review of "Linear EUS Accuracy in Preoperative Staging of Gastric Cancer: A Retrospective Multicenter Study"

_diagnostics, 2023, doi:10.3390/diagnostics13111842_

Round 1

Reviewer 1 Report

In this paper, the authors share their experience using linear EUS in preoperative gastric cancer staging. EUS is recommended by current guidelines for gastric cancer staging, especially for T assessment.

The article has pointed out several important aspects, and their results are according to similar studies. However, I have some recommendations:

Ø  First of all, the authors did not take into account the entire literature when comparing their results with similar articles. There is even a network meta-analysis published (PMID: 33467164) that I recommend being cited. Also line 36 to 39 require a citation.

Ø  In the results section, the histopathologic paragraph, there is no mention of the specific types of gastric cancer found. Perhaps you could also include a table along with the Lauren classification as well as the differentiation grade (G).

Ø  In the discussion section line 267 to 270, you mention that you could not perform EUS FNA and EUS FNB for lymph node involvement. There is no mention in this study in the material and methods that you intended to do so, or even if it s even recommend since you performed endoscopy with biopsy at first (this was mentioned in the study design). Please explain.

Ø  You mentioned as a limitation only the fact that is a retrospective study and that L-EUS was the main method to assess gastric cancer. However, the authors should be more accurate and should include other limitations such as gastric cancer characteristics (there is no clear relation of location if the gastroesophageal junction is included)

The article is well written, with minor editing required.

Reviewer 2 Report

This study aimed to evaluate the role of EUS vs CT for staging gastric cancer. The followings are my comments

1. Regarding CECT, what is the standard protocol of CECT in the study? Do patient received oral contrast? Do patients receive water as the contrast for better staging in your study? This point may explain the low accuracy of CECT in the study.

2. Line 268-270. Authors mention about the use of FNA/FNB for diagnosis LN. What is the indication of FNA/FNB for LN assessment in a gastric cancer patient? Would the assessment affect the surgery?

3. Please provide a reference for the sentence in line 45 “the use of L-EUS in GCstaging has increased … “

4. The study aimed to provide information of L-EUS in GC staging as compared with R-EUS. Please provide background information in the discussion ( i.e. in line 233 ) to let the authors know the difference. 

Round 2

Reviewer 1 Report

The authors fairly responded to my suggestions.

Minor editing is required.

Reviewer 2 Report

Dear Editor

 The authors response to questions raised. I have no more comments.